

# Development and primary validation of the School Health Assessment Tool for Primary Schools (SHAT-PS)

Maryam Kazemitabar[1,2], Danilo Garcia[3,4,5,6,7], JohnBosco C. Chukwuorji[8,9,10], Ricardo Sanmartín[11], Franco Lucchese[12,13], Kaveh Khoshnood[14] and Kevin M. Cloninger[6,15,16,17]

[1] Department of Psychology, University of Tehran, Tehran, Iran
[2] Promotion of Health and Innovation (PHI), International Network for Well-Being, Iran
[3] Department of Behavioral Sciences and Learning, Linköping University, Linköping, Sweden
[4] Centre for Ethics, Law and Mental Health (CELAM), Institute of Neuroscience and Physiology, University of Gothenburg, Gothenburg, Sweden
[5] Department of Psychology, Lund University, Lund, Sweden
[6] Department of Psychology, University of Gothenburg, Gothenburg, Sweden
[7] Promotion of Health and Innovation (PHI), International Network for Well-Being, Sweden
[8] Department of Psychology, University of Nigeria, Nsukka, Enugu, Nigeria
[9] Department of Psychology, College of Sciences and the Health Professions, Cleveland State University, Cleveland, OH, United States of America
[10] Promotion of Health and Innovation Lab (PHI), International Network for Well-Being, Nigeria
[11] Department of Developmental Psychology and Didactics, Faculty of Education, University of Alicante, Alicante, Spain
[12] Department of Dynamic and Clinical Psychology, and Health Studies, Sapienza University of Rome, Rome, Italy
[13] Promotion of Health and Innovation (PHI), International Network for Well-Being, Italy
[14] School of Public Health, Yale University, New Haven, CT, United States of America
[15] Anthropedia Foundation, St. Louis, MO, United States of America
[16] College for Public Health and Justice, Saint Louis University, St. Louis, MO, United States of America
[17] Promotion of Health and Innovation Lab (PHI), International Network for Well-Being, United States of Amercia

Corresponding authors
Maryam Kazemitabar,
maryam.kazemi64@ut.ac.ir
Danilo Garcia,
Danilo.garcia@icloud.com

## ABSTRACT

**Background.** School health programs need to target all aspects of physical, psychological, and social well-being. Using a slightly modified version of the COSMIN Risk of Bias checklist, we developed and conducted the first validation of the School Health Assessment Tool for Primary Schools (SHAT-PS).

**Method.** The exploratory sequential mixed method was used in this study. In the first phase, scientific databases were systematically searched to find school health models and instruments and 65 interviews were conducted with school stakeholders. The Colaizzi's method was used to code the qualitative data into themes. Then, a pool of items was created for each theme, rechecked by psychometric experts and then validated for content (*i.e.,* relevance, clarity, and comprehensiveness) by psychometric experts and individuals of the target population (*i.e.,* school personnel). In the second phase, classical test theory was utilized to analyze the validity and reliability of the resulting items from phase 1 among 400 individuals working at primary schools.

**Results.** The coding of the interviews resulted in ten themes that we labeled based on the theoretical literature: school health policies, community connections, health education, physical activity, health services, nutrition, psychological services, physical

environment, equipment and facilities, and school staff's health. The items created for each theme ended up in an initial pool of 76 items. In the final stage of phase 1, 69 items remained after the content validity assessment by experts and school personnel. In phase 2, the SHAT-PS items were tested using maximum likelihood exploratory factor analysis and confirmatory factor analysis. Of the 69 items from phase 1, 22 items were removed due to low factor loadings. The results showed that the 8-factor model was the best solution (chi-square/df = 2.41, CFI = .98, TLI = .97, RMSEA = .06). The discriminant and convergent validity of the SHAT-PS were evaluated as satisfactory and the scale had high internal consistency (Cronbach's alpha for all subscales > .93). The test-retest reliability was satisfactory—the intraclass correlation coefficient pooled was .95 (99% CI [.91–.98]). Moreover, the standard error of measurement resulted in an SEM pooled equal to 4.4. No discrepancy was found between subgroups of gender and subgroups of staffs' positions at schools.

**Conclusion**. The SHAT-PS is a valid and reliable tool that may facilitate school staff, stakeholders and researchers to evaluate the presence of the factors that promote health at primary schools. Nevertheless, in the process of validation, many of the items related to staff's health were eliminated due to poor factor loadings. Obviously, staff health is an important factor in the measurement of school health. Hence, we recommend that the validity and reliability of the SHAT-PS in other cultures should be done using the original 76-item version.

**Subjects** Global Health, Psychiatry and Psychology, Public Health, Science and Medical Education, Mental Health
**Keywords** School health, Psychometrics, Well-being, Factor analysis, COSMIN risk of bias checklist, Tool development, Reliability and validity, Assessment and evaluation

## INTRODUCTION

The World Health Organization (WHO) defines a health-promoting school as "a school that is constantly strengthening its capacity as a healthy setting for living, learning, and working" (*WHO, 2020*). The WHO's Global School Health Initiative was established in 1995 and endeavors to improve the health of students, school staff, families, and all the community members. In this context, school health programs have been led to several positive outcomes, such as improvement in academic achievement (*Ashok & Pathak, 2019*; *Minkkinen et al., 2017*) and physical activity (*McLoughlin et al., 2019*); the prevention of obesity and chronic diseases (*Centeio et al., 2018*); improvement in healthy behaviors (*Storey et al., 2016*) and mental health (*Dassanayake, Springett & Shewring, 2017*); healthy eating (*Graham et al., 2008*) and decrease in bullying and victimization (*Lewis et al., 2015*).

In other words, school health programs target all aspects of physical, psychological, and social well-being. School health promotion programs have been widely studied across the world. There are a wide variety of programs, but the models are mostly identical. For instance, the Focusing Resources on Effective School Health program, in partnership with UNESCO, UNICEF, the World Bank, WHO, and Education International, aims to improve learning outcomes by promoting health and good nutrition (*UNESCO. FRESH, 2020*); the Coordinated School Health Program, introduced by the U.S. Centers for Disease Control

and Prevention (CDC) since 1987, aims to improve health and academic achievement (*King & Lederer, 2019*); the Whole School, Whole Community, Whole Child model developed by the CDC and Association for Supervision and Curriculum Development aims to ensure keeping children healthy, safe, engaged, supported, and challenged (*Center for Disease Control and Prevention, 2020*). All of these models and approaches include the following school health factors: health education, physical education and physical activity, community involvement, nutrition, physical environment, health services, psychological and counseling services, employee wellness, family engagement, and school health policies (*Kazemitabar et al., 2020*). Although there is research on the factors that promote school health, there is little to no research on the psychometric properties of measures examining the prevalence of health factors in the literature. This gap is concerning given that children and teachers who struggle with illness and ill-being will be ill-equipped to benefit from the work of education. Having good measures of school health is an essential cornerstone in promoting discussion around issues of ill-being in schools and the consequences this has in the community and the society-at-large. Moreover, having a good understanding of the factors which promote or prevent school health is essential for effective health promotion and will help increase the knowledge-base for stakeholders such as health specialists at schools, school nurses, school principals, teachers, parents, policymakers, and the pupils themselves (*Kazemitabar et al., 2020*).

Research on school health promotion factors and instruments to evaluate school health do exist (*e.g., Andrews & Conte, 2005*; *Brener, Kann & Smith, 2003*; *Lee et al., 2014*; *Pinto et al., 2016*; *Weintraub & Erez, 2009*), but a recent systematic review showed that there are too few measures, and those measures are of poor psychometric quality (*Kazemitabar et al., 2020*). As the matter of fact, few studies did examine the psychometric properties of the scales. In the same review article, Kazemitabar and colleagues (*Kazemitabar et al., 2020*) implemented a slightly modified version of the COSMIN Risk of Bias checklist (*Prinsen et al., 2018*), which includes the following parameters: "tool development"[1], "content validity", "construct validity", "internal consistency", "cross-cultural validity/measurement invariance", "reliability", "measurement error", and "hypothesis testing for construct validity". The checklist highlighted several methodological errors in the existing literature. For example, none of the studies assessed measurement error and only one assessed construct validity. Additionally, most of the studies reported *Cronbach's alpha* as a reliability measure, despite the fact that it is a measure of internal consistency. Last but not the least, there was a lack of cross-cultural validity/measurement invariance and test-retest reliability. Hence, as showed by Kazemitabar and colleagues (*2020*), the COSMIN Risk Bias checklist can be used as a manual for the selection of both suitable statistical methods for tool development and the evaluation of measurement properties. Moreover, Kazemitabar and colleagues' systematic review and other research identified the need for the development of cross-cultural school health measures; which require appropriate adaptation and psychometric validation processes (*Arafat et al., 2016*).

In Iran, for instance, the school health approach has been applied since 2010. The evaluation of Health School Promotion Programs, however, is still a challenging endeavor, since there are no tools available that are adapted to the underlying characteristics of the

[1] The "Tool development" substituted the term "Patient Reported Outcome Measure (PROM) development" from the original COSMIN checklist. Moreover, in order to target populations at schools, the term "patient" was changed to "students", "teachers", and "principals" (for more details see *Kazemitabar et al., 2020*).

country and that are validated for comparison between countries. Therefore, as a first step, the present study aimed to develop and validate a tool to measure school health for primary schools in Iran based on a combination of the WHO's Health-Promoting School programs and CDC's Whole School Whole Community Whole Child models. Importantly, since the components or parameters related to primary schools are different from those applicable for middle and high schools (*e.g.*, sexual, drug and alcohol, and pregnancy issues), it was necessary to develop an instrument that specifically measures school health in primary schools. Moreover, as suggested in Kazemitabar and colleagues' systematic review (*Kazemitabar et al., 2020*), the present study applied the COSMIN Risk of Bias checklist as a guide to develop and measure the psychometric properties of what we choose to refer to as the School Health Assessment Tool for Primary Schools (SHAT-PS). A valid and reliable SHAT-PS can be used by policymakers, researchers, and health providers interested in primary schools in order to find and resolve the deficiencies and issues related to important factors that promote health among pupils and staff at primary schools.

## METHOD

### Ethical statement

This study received approval from the Ministry of Education of Iran (document number: 51241/64) and the University of Tehran, Faculty of Psychology and Educational Sciences. All participants obtained and signed a written consent form and were informed about the purpose of the study, they were assured of anonymity and also notified that they could withdraw from the study whenever they desired.

### Overview of the procedure

Exploratory sequential mixed method, that is, a combination of qualitative (phase 1) and quantitative (phase 2) research methodology was used in this study. In phase 1, a systematic search was conducted in different scientific databases and used the phenomenological approach to collect qualitative data (*i.e.*, interviews among health experts, principals, teachers, and staff of primary schools). These data were analyzed using Colaizzi's method in order to find, understand, describe, and depict the experiences found in interviews as well as emergent themes and their interwoven relationships. Then, the items were created using the codes of the themes and their content validity was systematically assessed by four experts and ten individuals from the target population. In phase 2, classical test theory was used to analyze the resulting items from phase 1 among primary school staff. Intraclass correlation coefficients were measured to assess test-retest reliability in a sub-sample answering to the SHAT-PS two weeks apart. In the following, the procedures used are described in detail, including each of the stages addressing how the SHAT-PS was developed and validated within each phase (see Fig. 1 for an overview of the development process behind the SHAT-PS and Fig. 2 for the COSMIN taxonomy).

*Phase 1: Development and content validity of SHAT-PS*

*Stage 1: Tool development.* The goal of this stage was to generate straightforward and unbiased items that measured factors promoting school health in primary schools. The
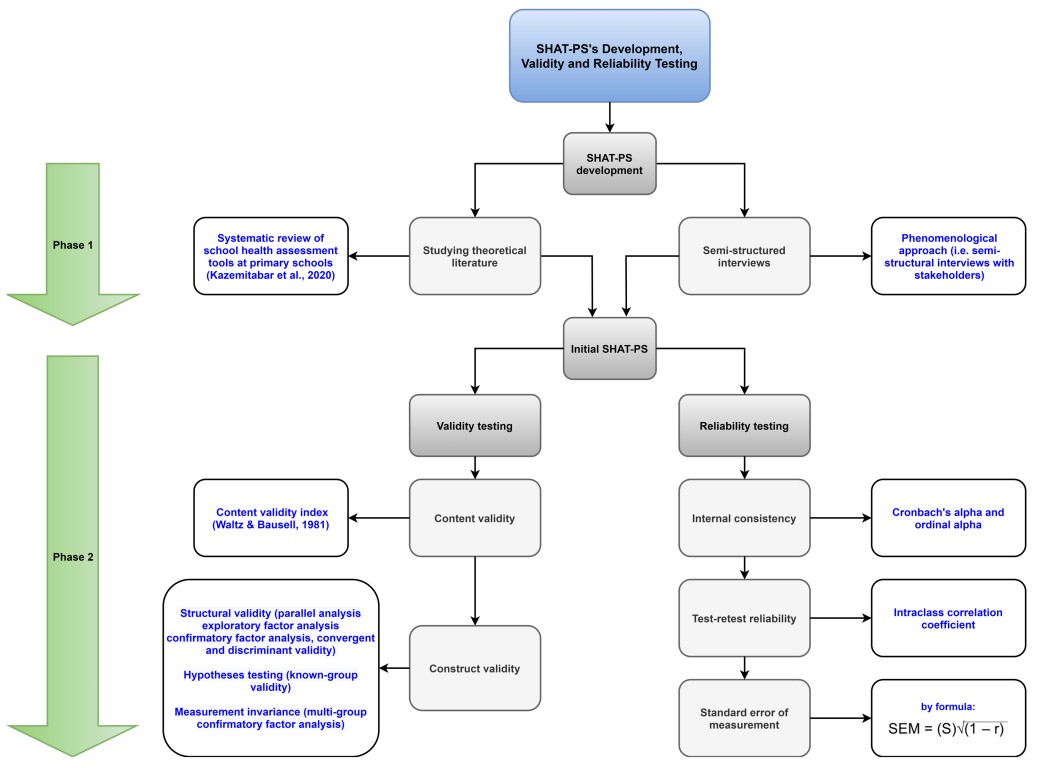

**Figure 1** Flowchart of the SHAT-PS development process.

items were generated through two sources: a) studying the theoretical literature of school health and b) interviewing primary schools' health experts, principals, teachers, and school nurses/health providers.

*Stage 2: Content validity.* Content validity index was employed to determine the comprehensiveness, relevance, and clarity of the items using a panel of experts.

## Phase 2: Examination of the SHAT-PS's validity and reliability

*Stage 1: Testing for validity.* The majority of the analyses from the COSMIN Checklist that are necessary to investigate the validity of an instrument were performed. Analyses used included construct validity of the instrument (*i.e.,* structural validity, measurement invariance, and hypotheses testing). In this study, the assumptions for exploratory factor analysis, such as parallel analysis, scree plot, eigenvalue higher than 1, Kaiser–Meyer–Olkin, and Bartlett's test of sphericity were tested. Then, exploratory factor analysis with fit indices using maximum likelihood estimation was conducted (*Muthén & Muthén, 2012*). A total of 22 items with factor loadings less than .5 were removed at this step *Hulland, 1999*; *Truong & McColl, 2011*. Then, confirmatory factor analysis was conducted to test results obtained from the maximum likelihood exploratory factor analyses. These procedures resulted in 47 items that were included in the SHAT-PS.

The chi-square value, degrees of freedom (df), root mean square error of approximation (RMSEA), comparative fit index (CFI), Tucker-Lewis index (TLI), and standardized root

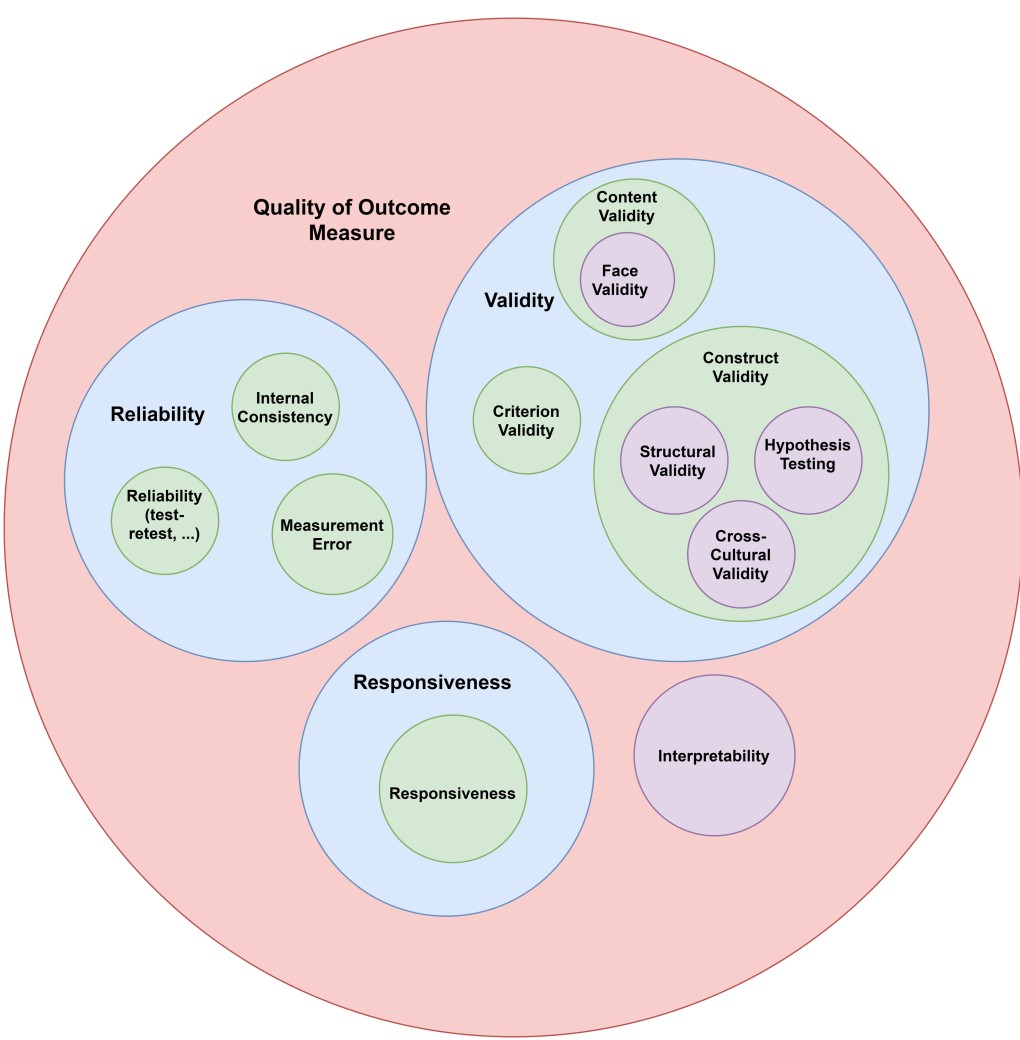

**Figure 2  COSMIN taxonomy (*Mokkink et al., 2010*).**

mean square residual (SRMR) were estimated and reported. Factor loadings for each latent variable plus measurement error, $t$-value, and $p$-value were reported. Measurement invariance of the construct was checked across gender groups. Convergent and divergent validity were estimated using the average variance extracted, maximum squared shared variance, and average squared shared variance (*Fornell & Larcker, 1981*). Finally, known-group validity was estimated to investigate whether the construct differentiates between subgroups of gender and school staff positions.

*Stage 2: Testing for reliability.* In this stage, the reliability was examined by looking at internal consistency, reliability (test-rest, inter-rater, or intra-rater), and standard error of measurement (*Mokkink et al., 2018*). The internal consistency of the factors was measured by Cronbach's alpha and composite reliability.

The test-retest method was also used to measure reliability. Here the intraclass correlation coefficient (ICC) was estimated for both test and retest administrations using the two-way

random effect model and then the ICC-pooled was calculated and reported. The test-retest interval was two weeks, which is considered long enough for participants to fail to recall their answers and short enough to certify stability. The test-retest respondents were a subgroup who volunteered to take the test for a second time two weeks apart from the first administration. The standard error of measurement (SEM) was estimated by a formula for each administration and finally, the SEM-pooled was calculated.

## Participants

In phase 1, the study population consisted of school health experts, principals, teachers, and staff at primary schools in Tehran. The sample size for interviews and panels was 65 individuals including four school health experts, 10 principals, 20 teachers, 15 staff, and 16 students. In phase 2, the sample size for tool administration needed to be at least seven times the number of items (cf. *Mokkink et al., 2018*). Forty-seven items remained after item screenings, and thus 400 respondents were considered sufficient for the psychometric study based on the COSMIN checklist criteria. Half of the participants (50%) were females and the other half (50%) were males. The participants' average age was 42.8 years old ($SD$ =4.61) with a range between 34 and 55. A total of 62 respondents, participated in test-retest administrations of the SHAT-PS to test reliability of the scale.

## Measures in phase 2
### School Health Assessment Tool for primary schools (SHAT-PS)

The SHAT-PS is a self-report measure designed to be distributed to primary school principals, teachers, and staff (*e.g.*, health experts, school nurses, school physicians, *etc.*). It is a 47-item tool measuring eight factors of school health policies (seven items), community connections (four items), health education (three items), physical activity (four items), health services (six items), nutrition (five items), psychological services (five items), and physical environment (13 items). The response format is based on a 4-point Likert scale including "*not at all*", "*a little*", "*quite a lot*", and "*very much*" and are scored from 1 ("*not at all*") to 4 ("*very much*"). A 4-point Likert scale was used in order to prevent neutral answers that may affect the validity of the results, in addition, it is difficult to semantically represent the idea of neutrality (*Asún, Rdz-Navarro & Alvarado, 2016*). None of the items has reversed scoring.

## RESULTS

### Phase 1: Tool development
#### Literature review and phenomenological approach

The databases of EBSCOhost, ProQuest, PubMed, Wiley, OpenGrey, and Google Scholar were systematically searched to find all school health models and instruments. School health's factors were extracted from the selected studies. Second, semi-structured interviews were performed with school health experts, and primary schools' principals, teachers, and staff in order to investigate any additional factors not present in the primary literature. The interview questions were validated by experts. Interviews continued until saturation was reached and there was no new information provided by the participants. A total of

65 individuals including five professors of educational psychology from the University of Tehran, five principals, 18 teachers, 17 school staff, and 20 students were selected from five areas in Tehran: north, south, east, west, and center. The sampling was purposive and the inclusion criteria were:

1- Participants except for students should have at least five years of experience in their job;

2- Participants except for students should have been employed by the school site for at least three years;

3- Students should be selected from the 4th, 5th, and 6th grades.

After selection of participants, each individual received informed consent forms and was made aware of the goals of study. Participants were also enlightened about how their personal information would be used and assured of anonymity.

The interviews were conducted and recorded, the recorded interviews were transcribed in MAXQDA 2020 software and studied and coded independently by two trained researchers. Colaizzi's method was used to code the qualitative data, which led to ten themes including school health policies, community connections, health education, physical activity, health services, nutrition, psychological services, physical environment, equipment and facilities, and school staff's health. The themes were named based on the theoretical literature review. Codes of the themes are displayed in Table 1. Then, items were created using the theme codes. For example, for the code "classroom lighting" of the theme "physical environment", the item "classroom lighting is sufficient" was devised. The items were rechecked by psychometric experts, and finally, an initial 76-item tool was devised. The 76-item version of SHAT-PS is provided in the Appendix.

### Content validity

Content validity index (*Waltz & Bausell, 1981*) was used to investigate the content validity of the initial 76 items. Four experts and ten individuals from the target population answered content validity index questions regarding relevance, clarity, and comprehensiveness of the items using a four-point scale (1 = not relevant/comprehensible/clear, 2 = somewhat relevant/comprehensible/clear, 3 = quite relevant/comprehensible/clear, and 4 = highly relevant/comprehensible/clear). The items with a content validity index of less than .70 were removed, those between .70 to .79 were revised and those higher than .79 were considered acceptable (*Zamanzadeh et al., 2015*). Seven items were removed as a result of content validity assessment and 69 items remained at this stage.

## Phase 2: Construct validity and reliability testing
## Structural validity
### Parallel analysis

Determining the number of retained factors in exploratory factor analysis (EFA) entails performing parallel analysis since the Kaiser criteria is not sufficient for this purpose (*Horn, 1965*). The parallel analysis was conducted by comparison between the actual eigenvalues and the average eigenvalues extracted from the syntax provided by Hayton and colleagues (*Hayton, Allen & Scarpello, 2004*). Then, the retained factors were those actual eigenvalues

**Table 1** Themes extracted from interviews and their codes.

| Themes | Codes |
|---|---|
| School health policies | Health education tools, monitoring camera, student-centered school, educational facilities, school amenities, single-shifted school, out-of-school activities, expert teachers for each lesson, token instead of money for students, talent identification, training for families and staff for student psychological and physical health, life skill training, using experienced staff. |
| Community connections | Media and TV cooperation, family engagement, Ministry of Education support, budget and funding, community engagement. |
| Health education | Course and syllabus for health education, health and hygiene culture, school nurse, student cooperation in school hygiene, health workshops and programs, enough health training hours per week. |
| Physical activity and education | Experienced physical education teacher, sport facilities, enough time for sports and play, sports competitions, teaching sports professionally. |
| Health services | Student health monitoring, health screening, student's health records, equipped room for school nurse, vaccination check |
| Nutrition | Education about healthy eating, lunchroom hygiene, attention to students' taste for meals, suitable buffet/restaurant, healthy and nutritious foods, safe and clean water. |
| Psychological services and counseling | Family parenting education, family economic problems, positive thinking education, addiction in family, parental conflict, psychological education to staff and parents, healthy and efficient interactions, student joy and happiness, calmness and tranquility for students, bullying control, psychological counseling for students, keep records for students' mental health, attention to students with special needs, happy climate at school, teachers' mental health, teachers' gentle interaction with students. |
| Physical environment | Hygiene and cleanliness of school, heating and cooling systems, joyful physical environment, proportional area of school and classroom to the number of students, the safety of school. |
| Equipment and facilities | Standardization of lighting, ventilation, desks, and chairs, *etc.*, double glazed windows, green space, equipped library and laboratory. |
| School staff's health | Staff's health, teachers' job satisfaction, adequate assistance staff (*e.g.,* teacher's assistants), sport facilities for staff, psychological counseling for staff, teachers' salary. |

higher than the average eigenvalues calculated. Finally, eight factors were considered valid for further analysis.

### Exploratory factor analysis (EFA) with goodness of fit indices

The EFA was conducted with an estimator permitting goodness of fit indices (like ML, *Truong & McColl, 2011*), for deciding whether a factor solution could be regarded as a

**Table 2  Summary of the model fit information.**

| Model | chi-square | df | p-value | CFI TLI | RSMEA SRMR |
|---|---|---|---|---|---|
| 1-factor | 16242.397 | 1034 | >0.001 | .311 .280 | .192 .191 |
| 2-factor | 12550.233 | 988 | >0.001 | .476 .427 | .171 .134 |
| 3-factor | 10174.097 | 943 | >0.001 | .582 .521 | .156 .108 |
| 4-factor | 8372.146 | 899 | >0.001 | .661 .593 | .144 .085 |
| 5-factor | 6305.625 | 856 | >0.001 | .753 .688 | .126 .077 |
| 6-factor | 4891.427 | 814 | >0.001 | .815 .755 | .112 .056 |
| 7-factor | 2980.151 | 773 | >0.001 | .900 .860 | .084 .034 |
| 8-factor | 1739.370 | 733 | >0.001 | .954 .933 | .059 .013 |

**Notes.**

df, degrees of freedom; CFI, comparative fit index; TLI, Tucker-Lewis index; RMSEA, root mean square error of approximation; SRMR, standardized root mean square residual.

special structural equation modeling case (*Brown, 2015*; p.26). The EFA was conducted in Mplus v7.4 through maximum likelihood estimation and oblique Geomin rotation. The results showed that the 8-factor solution had the best fit for the data. These factors have eigenvalues greater than 1 (Kaiser Criteria) and explained 83.36% of the total variance. Of the total 69 items, 22 items were removed due to low factor loadings. The EFA suggested seven items for factor 1 (school health policies), four items for factor 2 (community connections), three items for factor 3 (health education), four items for factor 4 (physical education and activity), six items for factor 5 (health services), five items for factor 6 (nutrition), five items for factor 7 (psychological services), and 13 items for factor 8 (physical environment). In addition, the Kaiser–Meyer–Olkin coefficient indicated that the sample size was adequate ($KMO = 0.939$) (*Kaiser, 1970*; *Kaiser, 1958*), and also Bartlett's test of sphericity (*Bartlett, 1951*) showed that the variables were sufficiently correlated (*chi-square* = 22139.573, *df* = 1081, $p < .001$). Table 2 shows the model fit information for 1 to 8-factor models. The indices of chi-square goodness of fit statistic, comparative fit index (CFI), Tucker-Lewis index (TLI), the root mean square error of measurement (RMSEA), and the standardized root mean square residual (SRMR) were all estimated and compared for all 8 models. All the fit indices suggested that the 8-factor solution is the best fit to the observed data ($\chi^2 = 1739.370$, *df* = 733, $p < .001$, CFI = .954, TLI = .933, RMSEA = .059, SRMR = .013). See also Fig. 3.

Moreover, the indices of $\chi^2/df = 2.37$ were less than 3 (*Kline, 2015*), CFI = .95, TLI = .933, RMSEA = .059, and SRMR = .013, thus, indicating a good fit (*Hooper, Coughlan & Mullen, 2008*; *Hu & Bentler, 1999*). The RMSEA was very close to the cut point of RMSEA

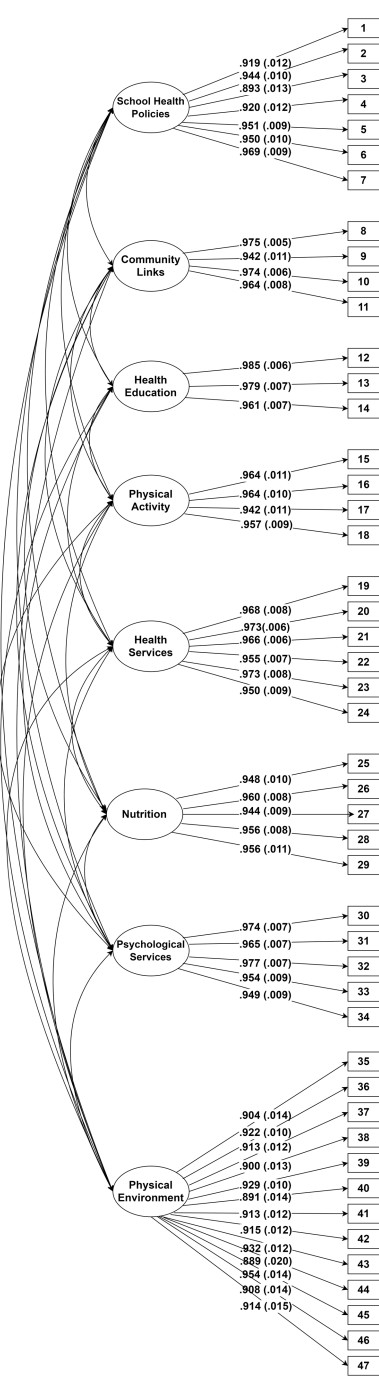

**Figure 3** Path diagram of final confirmatory factor analysis model for school health assessment tool for primary schools.

**Table 3  Pearson correlation coefficients between the factors in the SHAT-PS and the whole construct.**

| Factors | 1 | 2 | 3 | 4 | 5 | 6 | 7 | 8 | 9 |
|---|---|---|---|---|---|---|---|---|---|
| School health policy (1) | 1 | | | | | | | | |
| Community connections (2) | .184** | 1 | | | | | | | |
| Health education (3) | .377** | .019 | 1 | | | | | | |
| Physical education and activity (4) | .411** | .051 | .108* | 1 | | | | | |
| Health services (5) | .509** | .271** | .288** | .198** | 1 | | | | |
| Nutrition (6) | .358** | .209** | .368** | .312** | .060 | 1 | | | |
| Psychological services (7) | .456** | .118* | .124* | .228** | .444** | .135** | 1 | | |
| Physical environment (8) | .458** | .113* | .308** | .377** | .311** | .313** | .118* | 1 | |
| School health construct (9) | .806** | .357** | .499** | .548** | .641** | .544** | .505** | .754** | 1 |

$< .059$ (*Hu & Bentler, 1999*). The results from ML-EFA also showed that the 8-factor model had the best fit to the data.

### Confirmatory factor analysis (CFA)

To verify these results, a confirmatory factor analysis with weighted least squares –mean and variance adjusted estimation method (WLSMV; Mplus version 7.4) was conducted. The WLSMV estimation method was used due to the ordinal nature of our data (*Li, 2016*; *DiStefano & Morgan, 2014*). The CFA fit indices showed again a good fit for the 8-factor model (chi-square $= 2432$, $df = 1006$, chi-square/$df = 2.41$, CFI $= .98$, TLI $=.97$, RMSEA $= .06$, WRMR $=1.22$). A Pearson correlation between factors showed that most of the factors were correlated to each other and that they had strong correlations with the whole construct (Table 3). Table 4 shows the standardized factor loading and $t$-value for each item in the 8-factor model, which ranged from .88 to .98 and from 44.708 to 178.740 respectively. This shows that the items had high correlations to their factors. Moreover, the t-values were higher than 2.33 with $p < .01$ (*Bartlett, 1951*), hence, the items can be assumed to measure a common construct.

### Convergent and discriminant validity

As a means to verify the convergent validity of a scale, *Fornell & Larcker (1981)* suggest that the average variance extracted (AVE) should be greater than .5 and lower than the composite reliability (CR). In addition, to confirm the discriminant validity of the scale, the AVE should be greater than the maximum shared variance (MSV) and the average squared shared variance (ASV). Table 5 shows the values estimated for CR, AVE, MSV, and ASV for the 8-factor solution of the SHAT-PS, which suggest that the SHAT-PS items have the required correlation with the latent variables (*i.e.*, good convergent validity) and that the latent variables are distinctly different, thus, suggesting good discriminant validity (cf. *Adedeji, Lawan & Sidique, 2017*; *Zait & Bertea, 2011*). More specifically, the values of AVE for all the factors are higher than .5 and the CRs are higher than AVEs, therefore, the convergence validity of the instrument was considered acceptable. Besides, the AVEs are higher than the MSVs and ASVs, hence, the discriminant validity was also confirmed.

**Table 4  Factor loadings and t-values for the items in the SHAT-PS.**

| Factors and items | Loading | t-value |
|---|---|---|
| School health policies | | |
| 1- The school has specific rules for the rights and duties of individuals. | .919 | 76.641 |
| 2- There is a health organization at school (such as health care providers, health pioneers, and health promoters). | .944 | 94.433 |
| 3- School's educational facilities are adequate. | .893 | 70.715 |
| 4- School's amenities are adequate. | .920 | 78.400 |
| 5- School has the necessary facilities in case of unexpected events (such as fire, earthquake, *etc.*). | .951 | 105.598 |
| 6- School has regular programs for out-of-school recreational and educational activities for students. | .950 | 98.999 |
| 7- School staff are satisfied with their job. | .969 | 108.715 |
| Community connections | | |
| 8- Families work with the school to improve students' health. | .975 | 178.740 |
| 9- Supportive and charitable organizations work with the school to promote school health. | .942 | 84.987 |
| 10- Media and TV provide educational programs to promote school health. | .974 | 166.555 |
| 11- The Ministry and the Department of Education allocate sufficient budget and funds to schools. | .964 | 125.250 |
| Health education | | |
| 12- School organizes health-related educational workshops, conventions, and programs for students. | .985 | 166.473 |
| 13- Health educational posters, stands, and boards have been installed in classrooms, hallways, and halls. | .979 | 140.966 |
| 14- First aid training is provided to students. | .961 | 129.928 |
| Physical activity | | |
| 15- At least one sport is taught professionally at school, such as volleyball, basketball, handball, football, and so on. | .964 | 87.634 |
| 16- Sports competitions are held at the school. | .964 | 95.383 |
| 17- Adequate hours of the week are devoted to sport at school. | .942 | 82.810 |
| 18- School has enough sport facilities. | .957 | 105.955 |
| Health services | | |
| 19- There is a skilled nurse at school. | .968 | 125.388 |
| 20- Students' health status is assessed and registered in their health records. | .973 | 171.674 |
| 21- There are enough first aid kits for students and school staff. | .966 | 149.167 |
| 22- School nurses provide emergency health services to students and school staff. | .955 | 129.417 |
| 23- Teachers share students' health/educational issues with their parents. | .973 | 126.613 |

**Table 4** (*continued*)

| Factors and items | Loading | *t*-value |
|---|---|---|
| 24- The vaccination status of students is checked. | .950 | 101.255 |
| Nutrition | | |
| 25- School meals are prepared according to hygienic principles. | .948 | 97.004 |
| 26- School offers healthy and nutritious foods. | .960 | 116.484 |
| 27- Adequate information about healthy eating is provided to students. | .944 | 101.005 |
| 28- Healthy and safe drinking water is available. | .956 | 114.282 |
| 29- Buffet hygiene is monitored. | .956 | 85.569 |
| Psychological services | | |
| 30- Students are encouraged to be active in learning the lessons. | .974 | 137.851 |
| 31- Bullying and violence among students are prevented. | .965 | 131.023 |
| 32- Students with special problems (behavioral and learning) are identified and referred to the relevant specialists. | .977 | 148.340 |
| 33- Necessary psychological training is provided to school staff and parents. | .954 | 106.277 |
| 34- Students enjoy attending school. | .949 | 107.527 |
| Physical environment | | |
| 35- Restrooms are clean. | .904 | 63.640 |
| 36- The number of restrooms is sufficient. | .922 | 89.032 |
| 37- Drinking fountains/water coolers are clean. | .913 | 78.920 |
| 38- The number of drinking fountains/water coolers is sufficient. | .900 | 71.599 |
| 39- Classroom lighting is sufficient. | .929 | 89.014 |
| 40- The heating system in the classrooms is appropriate and sufficient. | .891 | 61.519 |
| 41- The cooling system in the classrooms is appropriate and sufficient. | .913 | 76.431 |
| 42- School is regularly inspected for the safety of buildings, windows, and equipment. | .915 | 77.447 |
| 43- Classroom desks and chairs are standard and comfortable. | .932 | 80.085 |
| 44- The area of the school is proportional to the number of students. | .889 | 44.708 |
| 45- The area of the classrooms is proportional to the number of students. | .954 | 66.651 |
| 46- School's physical environment is happy (colors, decorations, layouts, *etc.*). | .908 | 63.270 |
| 47- There is sufficient green space at school. | .914 | 62.561 |

**Notes.**
The Persian version of the instrument will be available by request through email to the corresponding authors.

**Table 5  The values for CR, AVE, MSV, and ASV for the 8-factor solution of the SHAT-PS.**

| Factors | CR | AVE | MSV | ASV |
|---|---|---|---|---|
| School health policies | .962 | .781 | .276 | .180 |
| Community connections | .955 | .842 | .081 | .029 |
| Health education | .955 | .877 | .158 | .076 |
| Physical education and activity | .931 | .771 | .189 | .083 |
| Health services | .959 | .798 | .276 | .117 |
| Nutrition | .950 | .793 | .152 | .082 |
| Psychological services | .946 | .779 | .229 | .083 |
| Physical environment | .966 | .687 | .231 | .105 |

Notes.

CR, composite reliability; AVE, average variance extracted; MSV, maximum shared variance; ASV, average squared shared variance.

## Hypotheses testing
### Known-group validity

Gender and positions of the school staff were studied based on the hypothesis that there were no differences in terms of school health between subgroups. A $t$-test was used to measure differences between females and males and Levene's analysis of variance was employed to assess differences between principals, teachers, and school health personnel (*e.g.*, nurses). As expected, the results showed that there were no differences in relation to gender ($t = -.429$, $df = 398$, $p = .668$) or position (*Levene statistic* $= .699$, $p = .498$; ANOVA: $F = 1.097$, $p = .335$) in the eight factors of school health as measured by the SHAT-PS.

## Measurement invariance

Measurement invariance was examined using multi-group confirmatory factor analysis, and changes in goodness of fit were inspected across gender groups in three invariance models: configural, metric, and scalar. Both females and males were similar in different characteristics of age, position, and district that school was located. Table 6 shows that the difference between CFA and RMSEA in the different models was less than .01 and the comparison between the models yielded $p$-values higher than .05. Hence, the school health construct does not vary across gender.

## Relibility testing
### Internal consistency

Cronbach's alpha and ordinal alpha coefficients were calculated and reported for each subscale of the 8-factor solution independently. The Cronbach's alpha and ordinal alpha coefficients were .96 and .97 respectively for school health policy .95 and .97 for community connections, .95 and .98 for health education, .93 and .96 for physical education and activities, .95 and .97 for health services, .94 and .96 for nutrition, .94 and .97 for psychological services, and .96 and .97 for physical environment. The Cronbach's alpha and ordinal alpha for the whole SHAT-PS scale were .95 and .97, respectively. The estimated values for internal consistency affirm high internal reliability among subscales, that is, high correlations between items with their respective subscales.

**Table 6  Invariance testing of the SHAT-PS using configural, metric, and scalar models.**

| Model | chi-square | df | *p*-value | CFI | RMSEA |
|---|---|---|---|---|---|
| Configural | 4469.675 | 2012 | .0000 | .886 | .078 |
| Metric | 4503.263 | 2051 | .0000 | .886 | .077 |
| Scalar | 4557.663 | 2090 | .0000 | .886 | .077 |

| Models compared | chi-square | df | *p*-value |
|---|---|---|---|
| Metric against configural | 33.588 | 39 | .7147 |
| Scalar against configural | 87.987 | 78 | .2059 |
| Scalar against metric | 54.400 | 39 | .0516 |

**Notes.**

df, degrees of freedom; CFI, comparative fit index; RMSEA, root mean square error of measurement.

### Test-retest reliability

Test-retest reliability was measured using intraclass correlation coefficient (ICC) analysis and a two-way random effect model in order to examine the reliability of the SHAT-PS. Two administrations were conducted with two weeks intervals on the same participants under similar conditions. The sample size for test-retest reliability ($n = 62$) was estimated by PASS v15. The ICC coefficient for the first administration was .97 (99% CI [.95–.98]) and for the second administration was equal to .94 (99% CI [.91–.96]). The ICC-pooled was .95 (99% CI:.91-.98), thus, only a very small alteration was found across administrations.

### Standard error of measurement (SEM)

SEM is an index of reliability and is calculated by the following formula (*Harvill, 1991*):

$$\text{SEM} = S\sqrt{1-r}$$

In which $S$ is the standard deviation and $r$ is the ICC (*i.e.,* reliability). SEM for the first administration was 4.19 and 4.61 for the second administration. The SEM-pooled was 4.4 with .95 confidence interval, which means that the scores are probably (with 95% confidence) dispersed 4.4 scores around the true score in both administrations.

## DISCUSSION

In this study, we have developed and validated the School Health Assessment Tool for Primary Schools (SHAT-PS). Our findings indicated that the SHAT-PS proved to be a valid and highly reliable tool for the measurement of the prevalence of important school health factors at primary schools in the Iranian context.

Nevertheless, in the process of validation, many of the items in the scale related to staff's health were eliminated due to poor factor loadings. Obviously, staff health is an important factor in the measurement of school health. It is plausible that since this factor has been neglected in Iran (*Charoghchian Khorasani, Peyman & Yaghobi, 2019*) along with the strict criteria for validity testing, resulted in the omission of this important factor in the SHAT-PS. Indeed, *Kaldi & Asgari*'s (*2003*) research indicated that the majority of primary schools' teachers are not satisfied with their jobs. In other words, the factor is not irrelevant, it is rather more relevant than any of the other factors. As the matter of fact, all school staff

in this study had extremely low scores in these items and their responses had almost no variance at all.

Known-group validity findings were in line with our expectations, the SHAT-PS produced similar scores among different gender and position groups. In other words, the perception of the presence or absence of these important factors did not seem to be significantly influenced by either the school staff's gender or position at the school. Moreover, the examination of participants' responses to the items in the factor "psychological and counseling services", being mainly "*not at all*" or "*a little*", showed that there is a serious need for assigning psychologists at primary schools in Iran. Especially in light of the sensitivity for mental illness and behavioral disorders within this age group (*Eslamieh, 2008*); but also due to the effect of promoting good mental health and well-being at this age has for functioning later in adulthood. These findings support studies that emphasize the necessity for the availability of psychologists or consultants at primary schools (*Gh, Naghavi & Ali, 2011*).

One of the SHAT-PS's strengths is that it was developed specifically for primary schools by considering their particular needs and requisites to promoting health for this age group. We recommend researchers in other countries to evaluate the validity and reliability of the 76-item SHAT-PS version (see Appendix), rather than the 49-item version, if they intend to use this instrument in different cultural and economic settings, infrastructures, ethnic groups, *etc.* (*Beaton et al., 2000*; *Sperber, Devellis & Boehlecke, 1994*). The cross-cultural study of the SHAT-PS will definitely affect the selection of the appropriate items measuring school health and it will help to understand whether it is applicable in other countries with all these distinctions or not.

The frameworks and programs currently used to evaluate school health are still in their infancy. Most of them are looking at optimization of current school conditions to approach or exceed normative standards, but are not focused on developing more expansive visions of well-being in the 21st century. As we enter into a period of increasing difficulty related to climate change, pandemic conditions, social inequities, and global economic difficulties, it is important that we reconsider how we can help children face a future they have not experienced and could never anticipate based on our own experiences in childhood. This study represents a step in that direction. The SHAT-PS is more comprehensive than all current primary school instruments reviewed in this study, since we merged factors from different models and examined the psychometric properties more accurately and comprehensively. Future research will need to examine even more comprehensive views of health and wellness based on our understanding of biological functioning of the human brain and body, our understanding of psychosocial development from developmental and positive psychology, and explore more deeply the impact of pedagogical approaches on staff and student health. All of this will be necessary to confront the real challenges of the 21st century and improving health at schools.

### Limitations

The main issue in criterion validity evaluation of questionnaire-based measures is the general absence of gold standards. Certain measures that are considered gold standards
might not be able to reliably assess the actual phenomena (*Bellamy, 2014*). As stated in the introduction, there is a need for the development of cross-cultural school health measurement tools. The SHAT-PS was developed and tested in a Middle Eastern context.

## Implications for future studies

Iran is a country with a student population of about 15 million, about 8 million of those were primary school students in 2019 (*ISNA News Agency, 2021*). It was not until 2010–2011 academic year that Iran started working on a program called ''Health Promoting Schools'' as an agreement between the two Ministries of Health and Ministry of Education (*Ramezani et al., 2016*; *Zarei et al., 2017*). Iran has detailed and accurate executive orders, assessment processes[2], financial rules and regulations, checklists, guidelines, policy-making, implementation, auditing, and monitoring of health-promoting schools. Hence, the number of health-promoting schools has increased exponentially since the program's inception. The question is, however, has this initiative improved the quality of school health in participating schools? Reveweing school health in Iran implies that there are imporatnt issues that Iran's health-promoting schools are facing such as inadequate funding for these approaches, lack of school nurses, unclear roles and responsibilities, lack of coordination, lack of incentives, lack of clarity and awareness of the importance among school staff to include health promotion in the curriculum, lack of student involvement, bad communication, and lack of adequate role models (*e.g.*, *Ramezani et al., 2016*; *Zarei et al., 2017*). Hence, the school health situation in Iran is different from that of Western societies, for example.

At the same time, the promotion of school health is a global concern and one of the challenges of the 21st century, thus, Iran and other countries are in equal need of validated methods for the promotion of well-being and also the prevention, identification, and treatment of mental illness in different populations and situations. Similar to Iran, many countries have launched school health initiatives, but have not necessarily had much success in their implementation. In Italy, the Miur-Directorate General for students integration and participation has launched various activities and collaborations over the years, aimed at schools of all levels, to protect the right to health, raise awareness on prevention issues and promote correct styles of life. In Spain, the term Health Promotion and Education in Schools appeared in 1990 in the *Ley Orgánica General del Sistema Educativo* (LOGSE) (*Government of Spain, 1990)*. In this law, the Health Promotion and Education topic was a cross-curricular element that must be present in all the curricular areas. After this law, the *Ley Orgánica 2/2006 de Educación* (LOE) (*Government of Spain, 2006*) established that in all the educational stages the Health Promotion and Education topic must be worked in the areas of environmental knowledge, natural sciences, physical education, and citizenship education. As a result, Health Promotion and Education should be integrated with projects inside the school environment. In Nigeria, for example, the National School Health Policy was approved and adopted in 2006. It describes a 5-component model: (1) school health service, (2) skill based health education, (3) healthful school environment, (4) school feeding service, and (5) school, home, and community relationship (*Dania & Adebayo, 2019*). Unfortunately, much of the efforts related to school health have been limited to policy

[2]Health-promoting schools are categorized based on the quality of their services with stars.

directives without much work on the implementation of such programs. Investigations of the school health situation in Nigeria have shown that children in Nigerian primary schools were not provided with basic health examination services and pre-entrance medical examinations, thus, there was the absence of essential baseline health information. Routine medical examinations that would have identified deviations from normal and made early referrals possible are not conducted in Nigerian schools (*Ojugo, 2005*; *Ola & Oyeledun, 1999*; *Kuponiyi, Amoran & Kuponiyi, 2016*; *Ademokun, Osungbade & Obembe, 2014*). In sum, implementation of the National School Health Policy is still grossly suboptimal in Nigeria.

## CONCLUSIONS

In these various contexts, the challenges are similar, but still unique to the specific cultural, economic, and historical conditions. What is certain is that, the promotion of school health requires adequate funding, training for participants at every level, improved communication, strong communication among all parties involved, and an awareness of the importance of the initiative. Absent these crucial ingredients, school health programs promote school health in name only, with most programs remaining just an intention to try to do better regarding student health within the status quo or "business-as-usual" approaches to modern education. There is no shortcut, schools, communities, and governments have to put a high value on child health and work on their own health in order to be good models of the health they hope to see in their children. Our work on the SHAT-PS is merely a small step in that global direction, and much more work will need to be done to help schools achieve the lofty goal of health-promoting institutions, especially when we consider that health involves an integrated biopsychosocial model to be sustainable.

Ultimately, there is a need for the promotion of school health at a global level. This is one of the challenges of the 21st century. We need to work with the validation of methods for the prevention, identification, and treatment of mental illness in different populations and situations; but also for the promotion, validation, and maintenance of well-being. In this framework, although the SHAT-PS was developed in a Middle East context, it seems to mirror the most important factors behind school health. Future studies, however, need to validate the original items generated in the interviews in new cultural contexts and consider how to expand the models to account for the changes to lifestyle in the 21st century that current school conditions are not built for. As Plato said, "If a man neglects education, he walks lame to the end of his life."

### Ethical approval and consent to participate

This study received ethical approval and license from the Ministry of Education of Iran (document number: 51241/64) and the University of Tehran, Faculty of Psychology and Education. Written consent was obtained from all the participants for the study. The participants were informed about the aims of the study, they were assured of anonymity and also advised that they could withdraw from the study whenever they intended to.

All protocols and methods were conducted aligned and in accordance with the relevant guidelines and regulations.

## ACKNOWLEDGEMENTS

We kindly thank all the participants in this study including students, principals, teachers, and staff of the primary schools in various districts of Tehran who have shared their thoughts and experiences on the phenomena under study in interviews and also replied to the items of the instruments patiently and honestly.

### Funding

The authors received no funding for this work.

### Competing Interests

The authors declare there are no competing interests.

### Author Contributions

- Maryam Kazemitabar conceived and designed the experiments, performed the experiments, analyzed the data, prepared figures and/or tables, authored or reviewed drafts of the paper, and approved the final draft.
- Danilo Garcia performed the experiments, analyzed the data, prepared figures and/or tables, authored or reviewed drafts of the paper, and approved the final draft.
- JohnBosco C. Chukwuorji, Ricardo Sanmartín, Franco Lucchese, Kaveh Khoshnood and Kevin M. Cloninger analyzed the data, prepared figures and/or tables, authored or reviewed drafts of the paper, and approved the final draft.

### Human Ethics

The following information was supplied relating to ethical approvals (i.e., approving body and any reference numbers):

This study received ethical approval and license from the Ministry of Education of Iran (document number: 51241/64) and the University of Tehran, Faculty of Psychology and Education.

### Data Availability

The raw data used in this study is available in the Supplemental File.

### Supplemental Information

Supplemental information for this article can be found online at http://dx.doi.org/10.7717/peerj.12610#supplemental-information.

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
