# Peer review of "Development and primary validation of the School Health Assessment Tool for Primary Schools (SHAT-PS)"

_PeerJ, doi:10.7717/peerj.12610_

## Round 0.1 · original submission · Minor Revisions

Include a diagram to show different methods and techniques used in each stage.

Reviewer 2 has suggested that you cite specific references. You are welcome to add it/them if you believe they are relevant. However, you are not required to include these citations, and if you do not include them, this will not influence my decision.

Reviewer 1 ·

Basic reporting

a. Abstract : Adequately written
b. Introduction: Adequately written
c. Methodology: Kindly refer to the statement “Exclusion criteria were individuals with less than five years of job experience and those 240 who did not intend to participate in the research.” There is no need to mention it in the exclusion criteria as the authors had already mentioned it in the inclusion criteria as “at least five years of job experience is needed for inclusion.” It implies that those who had less than five years of experience will be excluded.
d. Why the authors keep five years of work experience for inclusion? Is it an arbitrary selection or on the basis of any evidence?
e. Any abbreviations used in table and figures need to be mentioned with expanded version as foot notes

Experimental design

Adequate and appropriate

Validity of the findings

Appropriate

Additional comments

None

Reviewer 2 ·

Basic reporting

It is a well-written article. I found it interesting. The research question is well mentioned based on the background. The references are adequate and updated.

Experimental design

It is novel research with a fundamental impact on the school health system in Iran.

Validity of the findings

The impact of the study is long-term and significant on the community of countries like Iran. Also, the nearby countries can utilize and adapt the same instrument and/or same procedure.

Additional comments

I was wondering to have a small para on the schooling system in Iran in Introduction so that readers can get an idea from the article. Secondly, authors could discuss from the below-mentioned paper on psychometric validation
Arafat SY, Chowdhury HR, Qusar MM, Hafez MA. Cross cultural adaptation & psychometric validation of research instruments: A methodological review. J Behav Health. 2016;5(3):129-36. doi: 10.5455/jbh.20160615121755

---

## Round 0.2 · accepted · Accept

You have addressed all the queries raised by the reviewers.